# Evaluation of Electron Induced Crosslinking of Masticated Natural Rubber at Different Temperatures

**DOI:** 10.3390/polym11081279

**Published:** 2019-07-31

**Authors:** Ying Huang, Uwe Gohs, Michael Thomas Müller, Carsten Zschech, Sven Wießner

**Affiliations:** 1Department of Reactive Processing, Institute of Polymer Materials, Leibniz-Institut für Polymerforschung Dresden e.V., Hohe Strasse 6, D-01069 Dresden, Germany; 2Institute of Materials Science, Technische Universität, D-01069 Dresden, Germany; 3Institute of Lightweight Engineering and Polymer Technology, Technische Universität Dresden, Holbeinstrasse 3, D-01307 Dresden, Germany

**Keywords:** irradiation, masticated NR, additive-free, crosslink density, rheology

## Abstract

In this work, natural rubber (NR) was masticated using an internal mixer to fit the requirements of reactive blending with polylactide and characterized by size exclusion chromatography (SEC), Fourier-transform infrared (FT-IR) spectroscopy and dynamic rheology measurements. Subsequently, the effect of elevated temperatures (25 °C, 80 °C, and 170 °C) on the electron beam (EB) induced crosslinking and degradation of masticated natural rubber (mNR) in a nitrogen atmosphere without adding crosslinking agents has been investigated. The sol gel investigation showed that the gel dose of mNR slightly increased with increasing irradiation temperature, which is also confirmed by the swelling test. The chain scission to crosslinking ratio (*G_s_*/*G_x_*) was found to be less than 1 for irradiated mNR at 25 °C and 80 °C, suggesting a dominating crosslinking behavior of mNR. However, a significant increase of *G_s_/G_x_* ratio (~1.12) was observed for mNR irradiated at 170 °C due to the enhanced thermal degradation behavior at high temperature. A remarkably improved elasticity (higher complex viscosity, higher storage modulus, and longer relaxation time) for EB modified mNR was demonstrated by dynamic rheological analysis. Particularly, the samples modified at higher temperatures represented more pronounced elasticity behavior which resulted from the higher number of branches and/or the longer branched chains.

## 1. Introduction

Natural rubber (NR) is a biobased elastomer that was originally derived from milky latex found in the sap of rubber trees, such as genera Hevea and Ficus [1]. NR is usually reinforced by compounding with nanoparticles or fibers to get substantial improvements in strength and stiffness [2,3]. The reinforced NR is extensively used in many applications and products because of its excellent adhesion, high tear strength, and good resilience. It is well known that the mechanical properties of rubbers can be strongly influenced by the vulcanization due to the incorporation of crosslinking structures [4]. NR is conventionally cured by the incorporation of sulfur or peroxide and accelerators [5,6,7]. In consideration of environmental protection, electron beam (EB) irradiation is a sustainable method for modifying the polymer’s structure and properties by introducing crosslink unites [8]. This irradiation technique is based on a free radical process and can be carried out quickly, under well-controlled conditions and in the absence of a crosslinking agent. Nowadays, the EB modification method is strongly developed to modify the crosslinking and mechanical properties of NR [9,10,11].

It should be emphasized that the properties of EB modified NR are controlled by two competing mechanisms, i.e., chain scission and crosslinking, which depends on the radiation conditions, such as gas atmosphere, the presence of crosslinking agent as well as molecular mass of the NR. Recently, the influence of irradiation on the mechanical properties of NR at room temperature in an air atmosphere was largely investigated. Smitthipong et al. [12] irradiated synthetic polyisoprene rubber at room temperature in an air atmosphere with a dose ranging from 40 to 160 kGy and successfully prepared weakly crosslinked polyisoprene. Xu et al. [13] verified that greater tensile strength and elongation at break can be observed for radiation crosslinked polyisoprene rubber in comparison to sulfur vulcanization. The EB irradiated standard Malaysian rubber sheets at room temperature in the presence of air demonstrated that the tensile strength modulus increased significantly with an increasing dose whereas a mild growth in elongation at break was observed [14]. To enhance the crosslinking efficiency of NR and increase the degree of crosslinking at a low dose, some crosslinking additives are usually introduced. As reported [13,15,16], the radiation crosslinking of NR was mainly performed in the presence of some polyfunctional additives (e.g., butyl acrylate, diethyleneglycol dimethacrylate, tetramethylolmethane tetraacrylate). The biosecurity of vulcanized NR can be significantly influenced by any residual toxic acrylates, which has a detrimental impact on its acceptability for applications. It is noteworthy that NR can be crosslinked by ionizing radiation without any use of multifunctional monomers due to its high unsaturation in the backbone.

Furthermore, the molecular mass directly influences the gel dose of irradiated polymers as it will greatly influence the polymer chain segment mobility. Nowadays, most of the research studies are concentrated on the NR, the average molecular mass (*M_n_*) of which amounted to ≥500,000 g/mol. The viscosity of such NR is too high and it restricts the chain mobility due to strong physical entanglements [17]. Therefore, NR with a lower molecular mass should show higher crosslinking efficiency after irradiation due to higher polymer chain segment mobility. To the best of our knowledge, little work has been reported so far on the EB induced crosslinking and degradation of masticated NR at different irradiation temperatures.

The purpose of this paper is to investigate the EB induced crosslinking and scission behavior of masticated NR (mNR) with low molecular mass. The masticated NR was obtained by thermos-mechanical treatment of virgin NR in an internal mixer. Afterwards, it was characterized by size exclusion chromatography (SEC), Fourier-transform infrared (FT-IR) spectroscopy and rheological measurements. The mNR is intended to be used as toughening component for PLA in our future research work. Therefore, the mNR satisfying the requirements (suitable viscosity and storage modulus ratio) of toughening polylactide was subsequently modified by EB treatment in nitrogen atmosphere. The dose values varied from 0 to 200 kGy and three irradiation temperatures of 25°C, 80 °C and 170 °C were adopted, respectively. Afterwards, the effects of EB induced chain scission and crosslinking of mNR were comprehensively analyzed in terms of the gel measurement, swelling test and viscoelastic rheology investigations.

## 2. Experimental Section

### 2.1. Materials

The natural rubber (NR pale crepe, *M_n_*: 628,000 g mol^−1^, *DI*: 1.83) was purchased from Weber & Schaer, Hamburg, Germany.

### 2.2. Preparation of Masticated NR

The virgin NR material was dried in a vacuum oven at 40 °C for ~12 h before mastication. The masticated NR samples were prepared by mixing virgin NR at 80 °C in a Haake internal mixer with a chamber volume of 50 cm^3^ and at a rotor speed of 90 rpm. Various mixing times (5 min, 10 min, 20 min, and 40 min) were adopted to evaluate their influence on the molecular mass, the rheological behavior, and the functionality. The masticated NR samples were characterized by size exclusion chromatography (SEC), dynamical rheology, and Fourier transform infrared (FT-IR) spectroscopy.

### 2.3. Electron Beam Treatment of Masticated NR

All EB treatments were carried out in a nitrogen atmosphere using an electron accelerator ELV-2 (Budker Institute of Nuclear Physics, Novosibirsk, Russia). The electron energy and the electron current were kept constant at 1.5 MeV and 4 mA, respectively. The variable parameters were the dose (0, 25, 50, 100, and 200 kGy) and the irradiation temperature (25 °C, 80 °C, and 170 °C). To avoid excessive heating during EB treatment, the total dose was applied in steps of 25 kGy per pass. The mNR treated with 0 kGy was used as the control sample, which underwent the same process but no EB treatment. The mNR discs (thickness: 2 mm, diameter: 25 mm) were prepared by compression molding under vacuum at 80 °C and a pressure of 30 MPa for 5 min, subsequently cooled to room temperature at a rate of 10 K/min. The mNR plates were dried in a vacuum at 40 °C for ~12 h before the EB treatment. The EB treatment of mNR was carried out in a special chamber which was filled with nitrogen to avoid oxidative degradation. Subsequently, the samples were cooled and then removed from the irradiation chamber. Afterwards, the mNR samples were stored at room temperature in air and detailed characteristic analyses and measurements were performed.

### 2.4. Characterization

#### 2.4.1. Size Exclusion Chromatography (SEC)

The molecular masses were measured by using a gradient high performance liquid chromatography (HPLC) Series 1100 system (Agilent Technologies Inc., Santa Clara, CA, USA). The instrument contains two columns (MIXED-B-LS, 300 mm × 7.5 mm, 10 µm pore diameter) (DAWN Heleos-II, Wyatt Technology Corporation, Santa Barbara, CA, USA). The eluent was 1,2,4-Trichlorobenzene (TCB, 40 mL, HPLC grade) stabilized with 0.1% 2,6-di-tert-butyl-4-methylphenol (BHT) at a flow rate of 1 mL/min. NR samples were previously dissolved in TCB at 80 °C for 2 h under stirring. Solutions were filtered through 0.2 μm stainless steel frits prior to injection in order to remove NR phases which cannot dissolve completely within 2 h.

#### 2.4.2. Fourier Transform Infrared (FT-IR) Spectroscopy

FT-IR spectra of virgin NR and mNR were recorded with a Bruker Vertex 80 V spectrometer (Bruker, Karlsruhe, BW, Germany) over the range of 4000–400 cm^−1^, with 4 cm^−1^ of resolution, and 100 scans. Baseline corrections, using the rubber band method, were applied on the spectra. Finally, all spectra were vector normalized using OPUS software version 6.5.

#### 2.4.3. Sol Gel Content Test

To evaluate the effect of EB treatment on mNR, gel measurements were performed. Approximately 200 mg of the sample were extracted in boiling xylol for 16 h to remove the soluble part of EB modified mNR. After the extraction, the samples were dried in vacuum to remove the liquid solvent until obtaining a constant mass. The gel content was calculated in percent according to the following Equation (1):(1)$$X_{c}=(m_{1}/m_{0})\times 100\;\%$$
here *X_c_* is the gel content, *m*_1_ is the final mass after extraction, and *m*_0_ is the initial mass of the sample.

#### 2.4.4. Crosslinking Density Determination

The cross-link density was determined by equilibrium solvent swelling measurements in toluene at room temperature. A circular piece of 2 mm thickness was prepared to swell in toluene for about 72 h to ensure the swelling had reached equilibrium and then dried for 24 h in an oven until obtaining a constant mass. The masses of initial, swollen, and dried samples were measured using an analytical balance. The crosslink density value (*ν*) has been calculated using the modified Flory–Rehner equation [18], shown in Equation (2).
(2)$$\nu =-\frac{1}{V_{s}}\times \frac{\ln(1-V_{r})+V_{r}+\chi \;{\left(V_{r}\right)}^{2}}{{\left(V_{r}\right)}^{1/3}-0.5\;V_{r}}.$$

In Equation (2), *V_s_* characterizes the molar volume of solvent (toluene: 106.2 cm^3^ mol^−1^), *χ* represents the polymer solvent interaction parameter (Huggins interaction parameter) and it amounts to 0.393 in this case [19], and *V_r_* is the volume fraction of polymer in the swollen network, which is expressed by the following equation:(3)$$V_{r}=\frac{1}{A_{r}+1}=\frac{1}{Q}.$$

In Equation (3), *A_r_* represents the ratio of the volume of absorbed solvent to that of polymer after swelling. Q is the degree of swelling.

#### 2.4.5. Rheological measurements

Dynamic rheology was performed on an ARES rheometer (Rheometrics Scientific, New Castle, DE, USA) using parallel plate geometry. The diameter of the plate amounted to 25 mm and the gap between the plates was 1.5 mm. The test temperature for all samples was kept at 190 °C under nitrogen atmosphere. The frequency sweeps were carried out to measure the linear viscoelastic properties for masticated NR and EB modified mNR samples. The frequency sweeps were performed in the range of 100 to 0.1 rad/s at a strain of 1% to measure within the linear viscoelastic region.

## 3. Results and Discussion

### 3.1. The Characterization of Masticated NR

#### 3.1.1. The Molecular Mass

The influence of mastication process on the molar mass of masticated NR samples was thoroughly analyzed by SEC measurements. Figure 1a shows the molecular mass distribution of masticated NR as a function of mastication time. As evident from the curve, the molecular mass distribution peak occurred at the higher molecular mass range for the virgin NR without any mastication. After shear and heat treatment, the results implied that the peak gradually shifted to lower values with increased mastication time. This can be better understood from the number average molar mass (*M_n_*), the mass (*M_w_*) average molar mass, and the dispersity index (*M_w_*/*M_n_*), shown in Figure 1b. The *M_n_* reduced tremendously after 5 min mastication and it amounted to about half of that of the virgin NR. This reduction of *M_n_* is attributed to the increased number of polymer chains which can be resulted from three effects: (a) the disentanglements of physical network in neat NR and/or the disassembly of some protein couplings; (b) the breakage of the backbone main chain of NR by shearing; (c) the chain breakage from thermo-oxidative degradation. Consequently, short molecular chains were formed from polymeric backbone chains during the shear-milling process as mastication time increased. The most marked reduction of *M_n_* was attained after 5 min mastication, whereas a mild tendency of decrease in *M_n_* appeared with continuous increase of mastication time. The short chains have higher polymer chain segment mobility and were more favorable for radical induced chemical reactions.

A similar phenomenon was observed for the *M_w_* values, which decreased significantly as mastication time increased. The experimental results of *M_n_* and *M_w_* showed that the main chain scission occurred on NR polymer chains during the mastication process. The main chain scission effect involved a random molar mass distribution along the polymer chain and much more short chains were generated during the shear treatment. As a consequence, the dispersity index (*M_w_*/*M_n_*) was not change by mastication within the range of experimental uncertainty (Figure 1b).

#### 3.1.2. The Viscosity and Modulus

Dynamic rheology was carried out to get information about the viscoelastic behavior of virgin and masticated NR samples. Figure 2a represents the absolute value of complex viscosity (|*η*^*^|) versus angular frequency (*ω*) for masticated NR samples treated with different mastication time. It clearly indicates that a significant decrease in viscosity of thermal and shear treated NR was observed and a less pronounced shear thinning behavior can be seen with increasing time. The shorter chains may contribute to less entanglements and less friction induced by chains and therefore, lower |*η*^*^| values. In Figure 2b, the storage modulus (*G′*) and loss modulus (*G″*) of virgin and masticated NR are plotted as a function of *ω*. Compared to virgin NR, the storage and loss modulus decreased dramatically for the masticated NR. The reduced dynamic modulus is due to the fact that the less entangled polymeric chains existed. The short chains formed by mastication required less relaxation time to unfold and resulted in a fast relaxation at low frequencies. Moreover, the crossover frequency defined as the frequency where *G′* and *G″* are equal significantly shifted to a higher frequency, suggesting a reduced molecular mass. Finally, no crossover frequency was observed for masticated NR for 40 min within the studied range of angular frequency, showing a viscos behavior. This shift tendency was consistent with the molar mass measurement (see Section 3.1.1), where both *M_n_* and *M_w_* gradually decreased with enhanced mastication time. The viscosity and storage modulus ratio of masticated NR to PLA can fulfill the requirements of reactive melt blending as the NR was masticated with 40 min [20]. The masticated NR is intended to be used as toughening component for PLA in our future research work and was named as mNR in our following description.

#### 3.1.3. FT-IR Spectra

The effect of mastication in an air atmosphere (thermal oxidation) on the chemical structure of NR can be monitored by FT-IR spectroscopy. Figure 3a shows the vector-normalized FT-IR spectra of the virgin NR and mNR (masticated for 40 min), and three representative regions, i.e., 1600–700 cm^−1^ (Figure 3b), 1800–1500 cm^−1^ (Figure 3c) and 3600–3000 cm^−1^ (Figure 3d) were chosen to analysis the functionality.

As reported [21], the chain reactions that are more affected by the thermal treatment are those related to the unsaturations in natural rubber, that is, the C=C–H stretching, wagging and twisting modes, appeared at 1650–1670, 800–850 cm^−1^, and 3010–3070 cm^−1^ spectral regions, respectively. It can be seen in Figure 3b that the intensity of infrared absorption peaks at 840 cm^−1^, which is assigned to double bonds (=C–H) wagging modes in the NR molecules, decreased for the masticated natural rubber sample. Moreover, the region 1500–1200 cm^−1^ exhibits several characteristic band absorption bands, at 1445 and 1375 cm^−1^ assigned to –CH_2_ deformation and –CH_3_ asymmetric deformation, respectively. As observed, these characteristic absorption bands decreased for NR masticated for 40 min. Meanwhile, the absorption band at 1800–1700 cm^−1^ and 3600–3200 cm^−1^ broadened slightly and these features are assigned to carbonyl group (C=O) stretching vibrations (Figure 3c) and hydroxyl group (O–H) stretching or –NH stretching (Figure 3d), respectively. These functional groups of the oxidative products might result from alcohol and hydroperoxides [22]. These phenomena implied that a certain degree of oxidative degradation occurred in the NR chain during mastication. Due to the effect of heat treatment and oxygen, molecular chains broken into short pieces and consequently reduced the absorption intensity of the double bond, secondary methyl group, and methyl group. The results are consistent with some reports about natural rubber’s degradation [23,24]. Furthermore, the absorption bands at 1240 cm^−1^ and 1160 cm^−1^ might be also assigned to some impurities in NR such as protein, lipid, amino acids, and peptide [25]. It is known that these substances are thermally unstable and thus tend to decompose, especially at high temperature. As a consequence, these two bands showed a significant reduction after 40 min’s mastication.

### 3.2. Sol Gel and Crosslinking Density Analyses

The EB treatment of mNR was performed in an N_2_ atmosphere at different temperatures of 25 °C, 80 °C, and 170 °C, respectively. The EB induced chemical reactions involve crosslinking, chain scission, and grafting or long chain branching. The gel content can be used to effectively characterize the crosslinking behavior of irradiated polymers. Thus, the gel contents were measured and are shown in Figure 4a. It was evident that the crosslinked structures can be easily obtained and ascribe to the more flexible chains and less entanglements of mNR. The gel content amounted to 19% at 50 kGy for an EB treatment at 25 °C and increased dramatically with increasing dose. As the absorbed energy increased, more free radicals were generated in the mNR, and consequently more crosslinks formed. At a dose of 200 kGy, a maximum gel content of 82% was obtained, indicating the formation of highly crosslinked mNR. Mondal et al. [10] investigated the effect of absorbed dose on the gel content of virgin NR with a number molecular mass of 371,000 g mol^−1^. The result showed that the gel content increased from ~15% and ~64% as absorbed dose increased from 50 to 200 kGy. In comparison with the virgin NR, much larger gel contents of mNR were obtained at the same dose, indicating the more pronounced crosslinking efficiency after the shortening of NR chains due to mastication. Moreover, when the irradiation temperature increased, a small reduction in gel content was observed which might be attributed to the slightly enhanced thermal degradation of mNR chains at high temperature which can be verified by the reduced storage modulus of control samples in the rheology analysis section.

The appreciable change in the molecular architecture caused by EB treatment can be effectively and accurately detected by the equilibrium swelling test at room temperature [26]. Nevertheless, it should be noted that the equilibrium swelling measurements are only valid for polymers predominantly tending to crosslinking. As we know, during irradiation natural rubber belongs to the group of crosslinking dominant polymers. It is generally accepted that the swelling is directly related to the crosslink density of a polymer chain network, with less solvent uptake or penetration into the material indicating higher crosslink density. The crosslink density or the concentration of elastically effective chains can be conveniently determined from the Flory–Rehner equation [17]. As presented in Figure 4b, the crosslink density values linearly increased with increasing dose up to 200 kGy for all irradiation temperatures. More free radicals can be generated by increasing the dose and subsequently they contributed to the further increased number of crosslinks. As compared at various irradiation temperatures, almost no influence of temperature on the crosslink density can be observed, suggesting a similar network density at a given dose. The temperature dependence of crosslink density seems to be in contrast to the temperature dependence of gel content at enhanced temperature (Figure 4a). It has been indicated that the crosslinks include the true chemical crosslinks and physical crosslinks, such as loops and chain entanglements [27]. This deviation can be explained by the additional physical crosslinks formed by the different degree of branchings. Considering the gel content result (Figure 4a), the similar network density might point out that the improved degree of branching with increasing irradiation temperature. As the thermal load increased, additional chain scission resulted from the main chain breaking and/or breaking of crosslinks can be expected. Furthermore, the gel dose (*D_g_*) can be calculated from the fitted straight line as a function of the dose (Figure 4b), and the x-intercept of the fitted line is *D_g_*. It amounted to ~(45 ± 5), ~(45 ± 5), and ~(46 ± 5) kGy for an EB treatment temperature of 25 °C, 80 °C, and 170 °C, respectively. Within the overall uncertainty, these values are in good agreement with those calculated from the Charlesby–Pinner plots (shown in subsequent Section 3.3).

### 3.3. Evaluation of G_x_ and G_s_

The EB induced crosslinking and degradation behavior of mNR can be characterized by crosslinking and chain scission *G* values. The Charlesby–Pinner equation [28], shown in Equation (4), was usually used to evaluate the competing chain scission (*G_s_*) and crosslinking (*G_x_*) processes.
(4)$$s+\sqrt{s}=\frac{4.82\times 10^{6}}{G_{x}\cdot M_{n,0}}*\frac{1}{D}+\frac{G_{s}}{2\cdot G_{x}},$$
where *s* is the soluble fraction, *D* presents the dose value in kGy and *M*_*n*,0_ is the initial number average molecular mass in g mol^−1^. *G_s_* and *G_x_* are defined as the number of chain scission and crosslinking reaction yields caused by the absorption of 100 eV of irradiation energy, respectively.

The dispersity index (*M*_*w*,0_/*M*_*n*,0_) and *M*_*n*,0_ of mNR were determined by SEC measurement (see Figure 1b) to check the applicability of the Charlesby–Pinner Equation (4) and to evaluate the crosslinking behavior of mNR. Equation (4) is only valid for polymers having a dispersity (*M_w_*_,0_/*M_n_*_,0_) of 2 [29]. Since the dispersity of mNR amounted to 1.86, it can be considered to roughly fulfil this basic assumption of the Charlesby–Pinner model. According to Equation (4), the *D_g_* can also be calculated for s = 1 (or s + s^0.5^ = 2) and it amounted to ~(42 ± 4), (46 ± 5), and (49 ± 5) kGy for mNR modified at 25 °C, 80 °C, and 170 °C, respectively. Within experimental uncertainty, the gel dose calculated from the Charlesby–Pinner plot (Figure 5) was found to be consistent with that of equilibrium swelling measurement (~50 kGy, in Section 3.3). The ratio of *G_s_* to *G_x_* can be calculated from Equation (4) by using the fitted parameters of linear function in Figure 5. The characteristic parameters of chain scission and crosslinking are summarized in Table 1. As observed, the *G_x_* values amounted to 1.33 ± 0.13 for mNR at an EB treatment of 25 °C, 80 °C, and 170 °C. At 25 °C, the *G_s_/G_x_* ratio amounted to 0.56 and indicated the dominating crosslinking behavior of mNR at room temperature. A significant increase of the *G_s_/G_x_* ratio was found for 80 and 170 °C due to the enhanced chain scission (increased *G_s_*) behavior. As a consequence, the radiation induced mechanism seems to transfer from the predominant crosslinking at 25 °C to the slightly dominating main chain scission reaction at 170 °C. The enhanced chain scission behavior might be explained due to the additional thermal degradation of mNR chains and/or crosslinked regions during thermal treatment.

### 3.4. Rheological Properties

Linear viscoelastic analysis has been proved to sensitively detect the existence of branched and crosslinked structures of polymeric chains [30]. Thus, dynamic rheological measurements were carried out for unmodified and EB modified mNR samples. In Figure 6a, |*η*^*^| as function of *ω* is shown for mNR samples treated at room temperature with dose ranging from 0 to 200 kGy. It can be seen that a significant enhancement in viscosity of irradiated mNR was observed with increasing dose. This increase of |*η*^*^| was due to the stronger entanglements between chains by introducing branching (*D* < *D_g_*) and/or crosslinking (*D* > *D*_g_). The shear thinning behavior became more pronounced for EB modified mNR, which is ascribed to the random orientation of branched chains and disentanglement of molecular chains at high shear rate. Remarkably, the maximum growth in |*η*^*^| occurred at a dose of 200 kGy. This can be explained by a high gel content (Figure 4a) and crosslink density (Figure 4b) resulted from the dominating crosslinking behavior of mNR (*G_s_/G_x_* = 0.56).

In Figure 6b, the storage modulus of mNR, control and irradiated mNRs are presented as a function of ω. The polymer chains of unmodified and EB modified mNR showed a fast relaxation especially in the low frequency range. Compared with the unmodified mNR (0 kGy), the *G′* of EB modified samples increased significantly with increasing dose within the investigated frequency range. It is known that rubber chains usually demonstrate a viscoelastic behavior (i.e., partly viscous and partly elastic). The crosslinking process generates intermolecular chemical bonds and produces a tri-dimensional network, making a rubber more elastic than viscous. Therefore, a distinct characteristic of elasticity enhancement was observed by the enhanced dynamic moduli, which is due to the physical and chemical entanglements formed by branching and/or crosslinking. The Cole–cole plot, where the imaginary part (*η″*, *η″* = *G′*/*ω*) of the complex viscosity (*η**) is plotted as function of the real part (*η′*, *η′* = *G″*/*ω*), is shown in Figure 6c. As we known, the Cole–cole plot provides information about the mechanism of relaxation and mean relaxation time [31]. As the irradiation existed, a drastic deviation from the control sample’s shape and a pronounced upturning behavior were observed at high viscosities with increasing dose. In this case, it is apparent that the EB modified mNR samples demonstrated a slower relaxation process and longer relaxation time. The mNR samples modified at 200 kGy showed monotonically increased values of *η″* with increasing *η′* resulted from its elastic solid-like behavior.

Usually, the polymer chain structure can be characterized based on the van Gurp-Palmen (vGP) plot, showing *δ* as a function of the absolute value of complex modulus (|*G**|) [32]. It has been conducted on our rheological data to investigate the different branching levels and chain topologies of the modified mNR. As illustrated in Figure 6d, the rheological data of unmodified and EB modified mNR samples resulted in a typical curve from linear polymers with a strong decreased δ with increasing of |*G**|. After irradiation, the modified samples presented a more complex rheological behavior. It can be seen that a significant change was detected in the vGP plots shape, suggesting a significant change in chain topologies and/or polydispersity with absorbed dose. As mentioned in Trinkle’s work [33], such vGP plot shape is related with star-type branched polymers. Therefore, the results of dynamical rheology were consistent with those of the evaluation of EB induced crosslinking and degradation of mNR. Meanwhile, it indicates that some branched structures formed during EB treatment. In addition, the vGP plot can be also applied to research the elastic and/or viscous dominance. As δ ≤ 45^0^, the predominating elasticity was clearly shown for EB modified mNRs (*D* ≥ 50 kGy), due to the strong entanglements from EB induced crosslinking effect (see 3.2. and 3.3. subsection).

To compare the viscoelastic behavior of EB modified mNR at various temperatures, the corresponding *G′* and loss modulus (*G″*) versus *ω* are shown in Figure 7. As observed from Figure 7a (25 °C), the experiments demonstrated that the *G′* behaves more sensitively to an absorbed dose than the *G″*, leading to *G′* growing faster than *G″* with the same dose. Consequently, a crossover frequency (*ω*_c_) where *G′* and *G″* are equal was observed for an EB treatment at 25 kGy. For the mNR and control sample, the storage modulus was less than the loss modulus in the entire frequency range measured, implying a liquid-like state. In the case of samples having a higher dose (≥ 50 kGy), the gap between *G′* and *G″* increased and crossover point cannot be observed, which could be ascribed to the formation of a chemical network by crosslinks significantly influencing the viscoelasticity of the material. Hence, the EB irradiated mNR showed a pronounced solid-like behavior and stronger melt elasticity. This result is consistent with the gel content test (Figure 4a), where the gel fraction improved greatly with increasing dose.

Surprisingly, the *ω*_c_ gradually shifted toward lower frequency values as the irradiation temperature increased from 25 °C to 170 °C. As reported [34], this shift demonstrated an increase in molecular weight and/or the addition of long chain branching. Intuitively, it is reasonable that the material containing higher gel structure (crosslinking) exhibits larger molecular weight or stronger elasticity. However, the effect of long-chain branching on rheological behavior should not be neglected. As we know, the chain scission, branching, and crosslinking can be produced simultaneously during the irradiation. No matter in the investigations of gel content and *G* values evaluation, the degree of branching was impossible to be characterized and observed. As showed in the *G* values evaluation subsection, the chain scission to crosslinking ratio (*G_s_/G_x_*) significantly increased from 0.56 to 1.12 with increasing temperature, implying the enhanced chain scission might result from the thermal degradation during EB treatment. The short chains with enhanced polymer chain segment mobility can improve the formation of grafting structures. Therefore, the enhanced rheology behavior for mNR irradiated at high temperature can be attributed to the more pronounced branching possibility.

To more deeply distinguish the chain entanglement and polymer networks between EB modified mNR samples at various temperatures, the Bird–Carreau viscosity model is used to characterize the shear rate dependence of viscosity [35]:(5)$$\eta =\eta_{0}{\left[1+{\left(\lambda \dot{\gamma }\right)}^{2}\right]}^{(n-1)/2},$$
where *η*_0_ is the zero shear viscosity, *λ* is the characteristic time, and *n* represents the power law index. The n value can be another important parameter providing information about the relaxation mechanism. The rheological viscosity curves of mNR irradiated at different temperatures were fitted by the Bird–Carreau viscosity model (Equation (5)). As a consequence, the shear-thinning power law index was characterized in terms of the dose values and shown in Figure 8. It is accepted that the n value can reflect the shear rate sensitivity of viscosity in the range of shear rates. Hence, the power law index is logically lowered with the increasing shear sensitivity. As presented in Figure 8, for all EB modified mNR, the power law indices were less than 1, and reduced with increasing absorbed dose. This phenomenon indicated that more obvious pseudoplastic or shear-thinning behavior can be detected as dose increased. Furthermore, it is apparent that the n value reduced more obviously with increasing dose as irradiation temperature increased. The reduction in n value can be explained in terms of greater numbers of interactions and entanglements restricting the individual chains motion. This power law index result is consistent with and also supported by the crossover frequency shifting phenomenon illustrated in Figure 7.

## 4. Conclusions

In this work, masticated NR was prepared by treating virgin NR in an internal mixer with different mastication times. Subsequently, the mNR with low molecular mass, reduced storage modulus, and enhanced polymer chain segment mobility was irradiated at different temperatures in a nitrogen atmosphere without adding crosslinking agents. The crosslinking behavior was studied by gel content measurements and equilibrium solvent swelling measurements. Within the overall uncertainty of both methods, the gel dose of mNR amounted to 45 kGy and was influenced by the irradiation temperature used. Using the Charlesby–Pinner equation, a constant value of *G_x_* (1.33 ± 0.13) was determined. As the irradiation temperature increased, the *G_s_/G_x_* ratio significantly enhanced. This might be due to the additional extended thermal treatment of crosslinked mNR with increasing irradiation temperature. The occurrence of branching and/or crosslinking structures was confirmed by dynamic rheological measurement, since the elasticity of irradiated mNR gradually improved with increased dose. Moreover, the samples modified at higher temperatures presented more pronounced elasticity behavior which might result from the more number of branches and/or the longer branched chains.

## Figures and Tables

**Figure 1 polymers-11-01279-f001:**
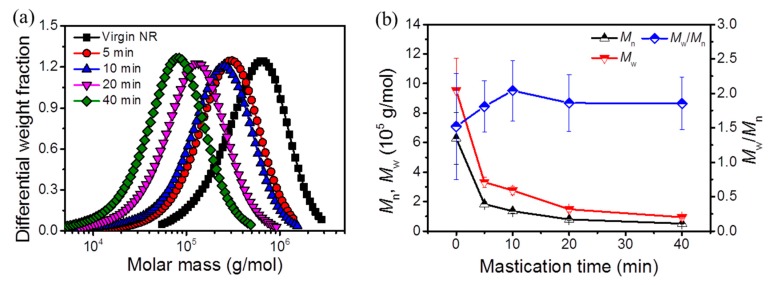
The molecular mass of masticated natural rubber (NR) with different mixing time: (**a**) Molecular mass distribution; (**b**) the number (*M_n_*), mass average molecular mass (*M_w_*) and the dispersity (*M_w_/M_n_*).

**Figure 2 polymers-11-01279-f002:**
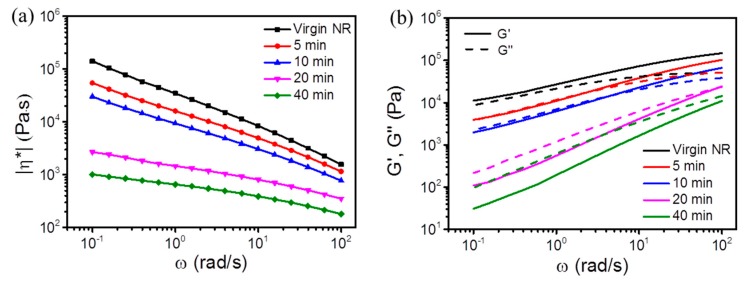
Variations of complex viscosity (**a**), storage modulus (**b**, straight line) and loss modulus (**b**, dashed line) as functions of angular frequency for masticated NR with various mixing time.

**Figure 3 polymers-11-01279-f003:**
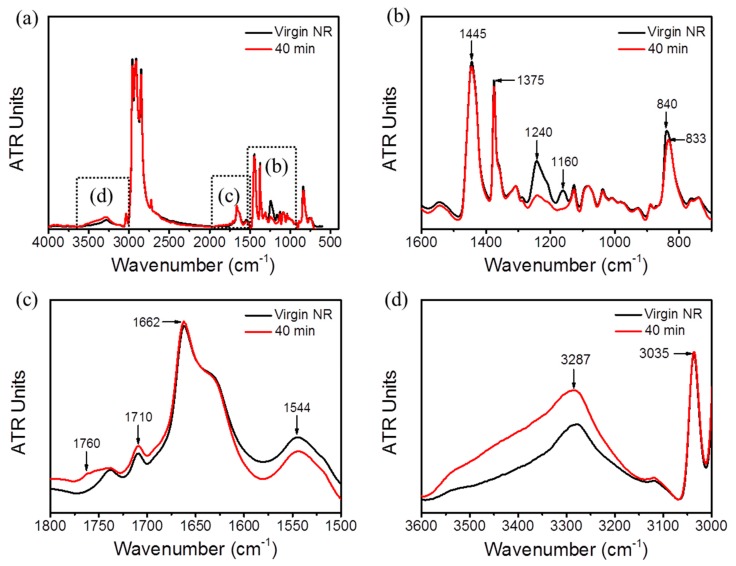
Evolution of vector-normalized FT-IR spectra of virgin NR and masticated natural rubber (mNR) recorded in different domains: (**a**) In the domain 4000–400 cm^−1^; (**b**) In the domain 1600–700 cm^−1^; (**c**) In the domain 1800–1500 cm^−1^; (**d**) In the domain 3600–3000 cm^−1^.

**Figure 4 polymers-11-01279-f004:**
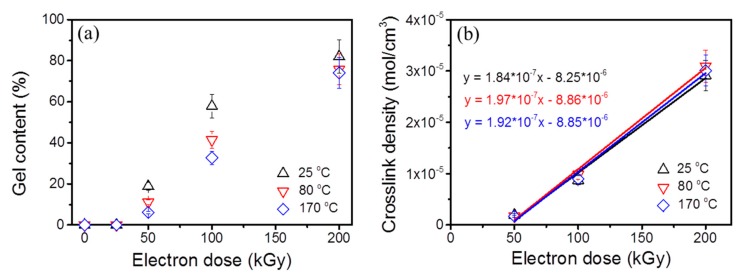
(**a**) Gel content of irradiated mNR with various applied temperatures and doses; (**b**) Variation of crosslink density with dose.

**Figure 5 polymers-11-01279-f005:**
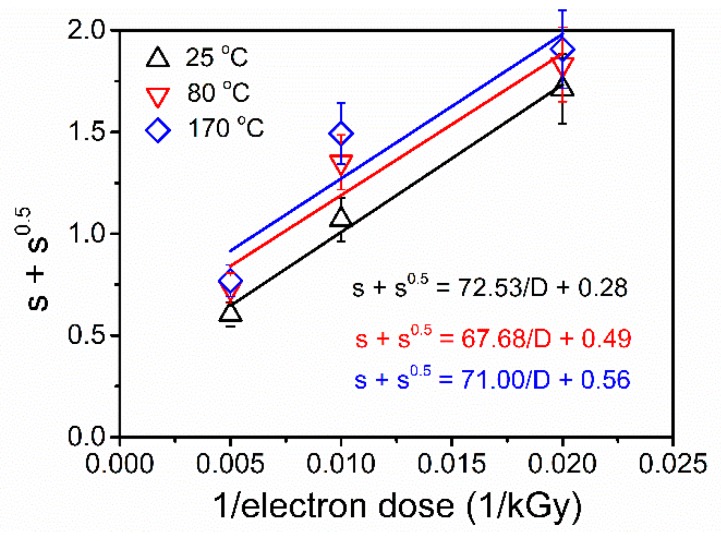
Charlesby–Pinner plots of mNR modified at 25, 80, and 170 °C with dose ranging from 0 to 200 kGy.

**Figure 6 polymers-11-01279-f006:**
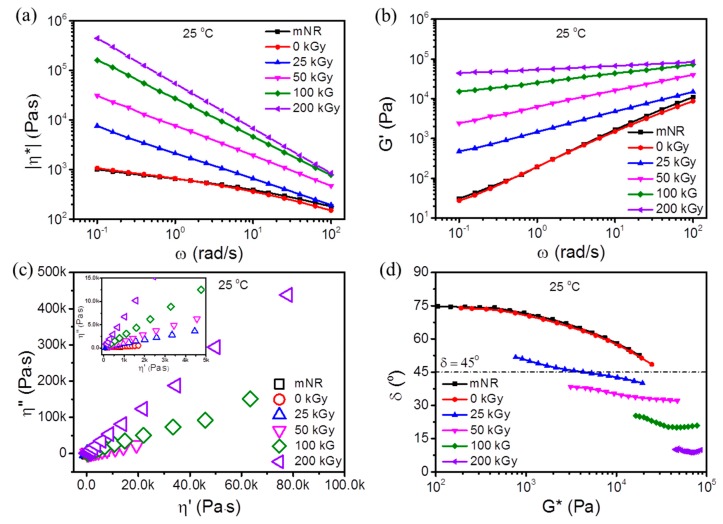
Rheology investigations: Effect of angular frequency on the complex viscosity (**a**) and storage modulus (**b**); Cole–cole plots (**c**) and van Gurpe-Palmen plots (**d**) for mNR samples EB modified at 25 °C.

**Figure 7 polymers-11-01279-f007:**
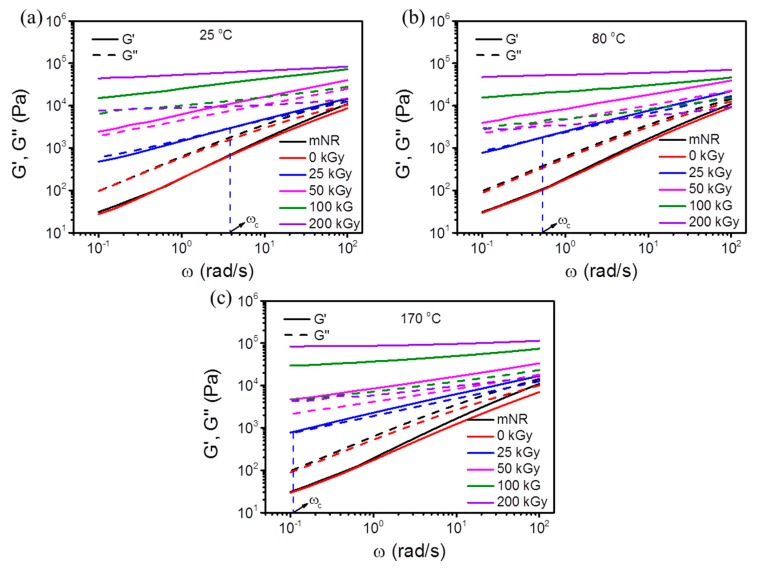
Storage modulus (a, b, c, solid line) and loss modulus (a, b, c, dashed line) as a function of angular frequency for mNR irradiated at 25 °C (**a**), 80 °C (**b**), and 170 °C (**c**) with the dose ranging from 0 to 200 kGy.

**Figure 8 polymers-11-01279-f008:**
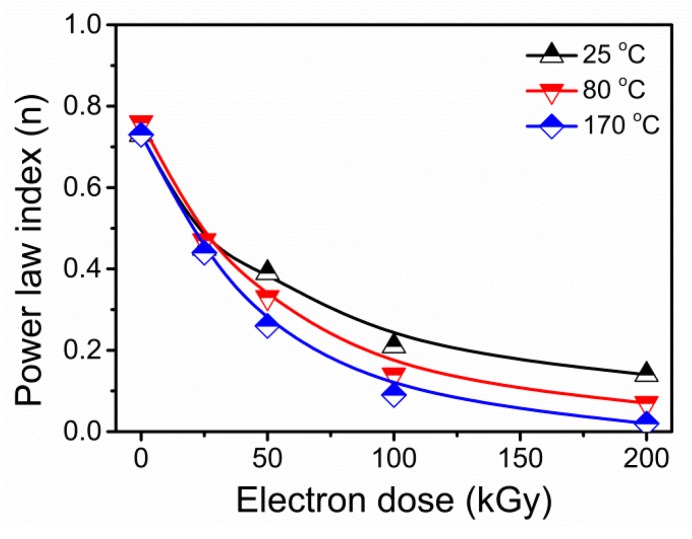
Power law index (n) for mNR irradiated at 25 °C, 80 °C, and 170 °C with the dose ranging from 0 to 200 kGy.

**Table 1 polymers-11-01279-t001:** G-values of crosslinking (G_x_) and scission (G_s_) of EB irradiated mNR at various temperatures.

Polymer	*M_n_**_,0_* (g/mol)	*T* (°C)	*a* (kGy)	*b*	*G_x_*	*G_s_*	*G_s_/G_x_*
mNR	51,000 ± 5000	25	72.53	0.28	1.30 ± 0.13	0.73 ± 0.07	0.56
		80	69.68	0.49	1.36 ± 0.13	1.33 ± 0.13	0.98
		170	71.00	0.56	1.33 ± 0.13	1.49 ± 0.15	1.12

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
