# Peer review of "Evaluation of Electron Induced Crosslinking of Masticated Natural Rubber at Different Temperatures"

_polymers, 2019, doi:10.3390/polym11081279_

Round 1

Reviewer 1 Report

In this manuscript, the authors are interested to study the EB crosslinking of masticated natural rubber at different temperatures. The scope of the paper is interesting because few studies (about NR) in the literature deal with this way of crosslinking.

About the manuscript itself:

In the Results and discussion part

Line 159-160 :The authors wrote “This reduction of Mn is attributed to the increased number of polymer chains, resulting from breakage of backbone main chain o NR by shearing”. The masticated NR samples being prepared by mixing virgin NR at 80°C, without taking into account the increase of temperature during mastication, the role of thermo-oxydation on polymer chains breakage can’t be neglected and has to be mentioned.As previously demonstrated by several authors, neat NR is a physical crosslinking network. Is the sharp drop of molecular weight at the beginning of the mastication mainly due to the breakage of this physical network?

In the Characterization part (line 114) the authors explain “Solutions were filtered through 0,2µm stainless frits prior to injection in order to remove the large insoluble particles”. What are these insoluble particles and what is their ratio in NR samples. If solutions have to be filtered before SEC experiments, how the authors explain that the gel content of NR samples is equal to zero at 0 and 25 kGy (Figure 4)?

The end of the discussion part (line 389- 401) is somewhat confuse and has to be rewrite for a sake of clarity.

From the previous remarks, this manuscript needs minor modifications before reaching the eligibility criteria for a publication in Polymers.

Reviewer 2 Report

In this manuscript, electron beam induced crosslinking and scission of masticated natural rubber is investigated using different characterization methods (SEC, FTIR, rheology). The motivation which is “satisfying the requirements for toughening polylactide”, is not clearly introduced, in my opinion.

The experiments are clearly described and the results supported by data. The manuscript is well presented, rather well written. Some clumsy sentences could be rewritten. Below some corrections/suggestions may be found.

Abstract line 20: a dominating crosslinking behavior in mNR

Line 59: It is

Line 65: has, restricted, choose tense

Line 68: add (mNR) to define in text.

Line 71: thermo-mechanical

Line 74: Better explain in introduction “The masticated NR satisfying the requirements of toughening polylactide was…” that is understandable only long later on lines 195-197. This is actually mentioned in exp part (EB treatment), not useful there.

Line 193: viscous behavior

Line 201: masticated for 40 min

Line 225: It is

Line 227: choose min or minutes (cf line 201)

Line 233: “It was evident that mNR tended to create crosslinked structures rather than degradation due to the presence of considerable amount of unsaturation”. Clumsy sentence. It is not mNR that creates, but mastication. Crosslinked structures versus degraded structures, not degradation. “due to …” at the end of the sentence is ambiguous.

Line 240: from 15 to 64

Line 273: uncertainty

Line 293: see 3.3 section IN 3.3 section

Line 304: Column Polymer non useful

Line331: as we know

Line 343: presented, not represented

Lines 347: “The results of dynamical rheology were consistent with that of the evaluation of EB induced crosslinking and degradation of mNR as well as confirmed the EB induced branching of mNR”. Long and clumsy sentence.

Line 370: It is

Line 371: exhibits, not behaves

Line 377: might result

Line 414: presented

Line 415: might result

To conclude, this manuscript is a good experimental paper that deserves publication, after minor improvements suggested above.
